
# Wave height return periods from combined measurement–model data: A Baltic Sea case study

Jan-Victor Björkqvist[1], Sander Rikka[2], Victor Alari[2], Aarne Männik[2], Laura Tuomi[1], and Heidi Pettersson[1]

[1]Finnish Meteorological Institute, Marine Research, Erik Palménin aukio 1, P.O. Box 503, FI-00101 Helsinki, Finland
[2]Tallinn University of Technology, Department of Marine Systems, Akadeemia tee 15a, 12611, Tallinn, Estonia

**Correspondence:** Jan-Victor Björkqvist (jan-victor.bjorkqvist@fmi.fi)

**Abstract.** This paper presents how to account for the lack of sampling variability in model data when they are combined with wave measurements. We addressed the dissimilarities between the types of data by either: i) low-pass filtering the observations or ii) adding synthetic sampling variability to the model. Measurement–model times series combined with these methods served as the basis for return period estimates of a high wave event in January 2019. During this storm northerly wind speeds in the Baltic Sea rose to 32.5 m s$^{-1}$ and an unprecedented significant wave height of 8.1 m was recorded in the Bothnian Sea sub-basin. Both methods successfully consolidated the combined time series but produced slightly different results: using low-pass filtered observations gave lower estimates for the return period than using model data with added sampling variability. Extremes in both types of data followed the same type of theoretical distributions, and our best estimate for the return period was 104 years (95 % confidence 39–323 years). A similar wave event can potentially be more likely in the future climate, and this aspect was discussed qualitatively.

## 1 Introduction

We have two fundamental ways to get wave information. Models cover large areas, recreate past events, and quantify impacts of future changes in forcing. Their major weakness is that they are not necessarily accurate enough for rare events, such as storms (Cavaleri, 2009). Measurements, again, can provide a certain ground truth that is unmatched even by the best of models. But point measurements can't confidently represent large areas – neither can they be made in the past or in the future. For certain purposes remote sensing products combine the versatility of (numerical) models and the reliability of (in situ) observations (e.g. Young et al., 2011; Salcedo-Castro et al., 2018). Often, however, models and measurements are used in combination; typically a model is validated and calibrated with observations (e.g. Bidlot et al., 2002; Haiden et al., 2018) before being used to extend limited measurements in time or space (e.g. Caires et al., 2006; Soomere et al., 2008; Breivik et al., 2013).

A lot, then, hinges on that instruments properly capture physical phenomena despite being limited by unavoidable sampling variability (Longuet-Higgins, 1952; Bitner-Gregersen and Magnusson, 2014). This uncertainty can be around 10 % in significant wave heights measured by wave buoys (Donelan and Pierson, 1983), and Forristall et al. (1996) determined it to lead to a



3–7 % bias when estimating 100 year wave heights. Also wave modellers have noted that their simulated extremes represent a mean over a few hours – not a single 20–30 minute measurement (Bidlot et al., 2002; Aarnes et al., 2012; Breivik et al., 2013).

The statistical behaviour of extremes is also unknown. Extreme value analysis has a solid – but wide – theoretical framework (e.g. Coles, 2001) leaving researchers to determine how to define extremes (e.g. Méndez et al., 2006; Orimolade et al., 2016) and with what distribution to model them (e.g. Aarnes et al., 2012; Haakenstad et al., 2020). In addition, there is a globally uneven trend in significant wave heights (Hemer et al., 2013), with the decreasing ice cover being a dominant factor in high latitudes (Stopa et al., 2016; Groll et al., 2017). All the aforementioned issues obscure estimates of how often to expect extreme wave events during, for example, the next decade.

This study examines how to best combine measured and modelled data and how well this merged data set is suited to analyse extremes. The specific focus is a record wave event that took place on 1 January 2019 in the seasonally ice-covered Baltic Sea. There exist no wave buoy data from the study location prior to 2011, but the observations were complemented by hindcasts covering 1965–2013 (Björkqvist et al., 2018; Tuomi et al., 2019). We consolidated the data to a continuous time series using two methods, both of which can be used universally. The return period of the wave event was then inferred from theoretical distributions that were fitted to the combined measurement–model data.

The paper has the following structure. Observations and model data are described in Sect. 2, while methods for combining data and fitting distributions are presented in Sect. 3. An overview of the storm is provided in Sect. 4, before estimating the return period of the wave event in Sect. 5. We discuss our results with respect to the current climate and a future declining ice cover, and end by concluding our findings.

## 2 Data

### 2.1 Wave buoy observations

Wave buoy data in the Bothian Sea were available from two sites. The Finnish Meteorological Institute (FMI) has moored a wave buoy in the eastern part of the Bothnian Sea (61° 48' N, 20° 14' E, Fig. 1) at a depth of 120 m. The data covers 79 % of the time 2011–2019, which is a high coverage considering that the wave buoy cannot measure during the ice season. The wave buoy at Finngrundet (60° 54' N, 18° 37' E, Fig. 1) in the south-western part of the Bothnian Sea is operated by the Swedish Meteorological and Hydrological Institute (SMHI). This buoy is moored at a depth of roughly 70 m about 10 km south-east of an underwater bank. Both wave buoys are Datawell Directional Waveriders. At the Bothnian Sea wave parameters were derived from the wave spectra available every 30 minutes on the buoy's memory card. The Finngrundet data for the storm duration were extracted from SMHI's open data portal (parameters available every hour).

The significant wave height was determined as $H_s = H_{m_0} = 4\sqrt{m_0}$, where $m_0$ is the integral of the wave spectrum. The peak period, $T_p$, was determined as the argument maximum of the spectrum, and the directional parameters of mean direction at the spectral peak, $\theta_p$, and the directional spreading was determined following e.g. Kuik et al. (1988).


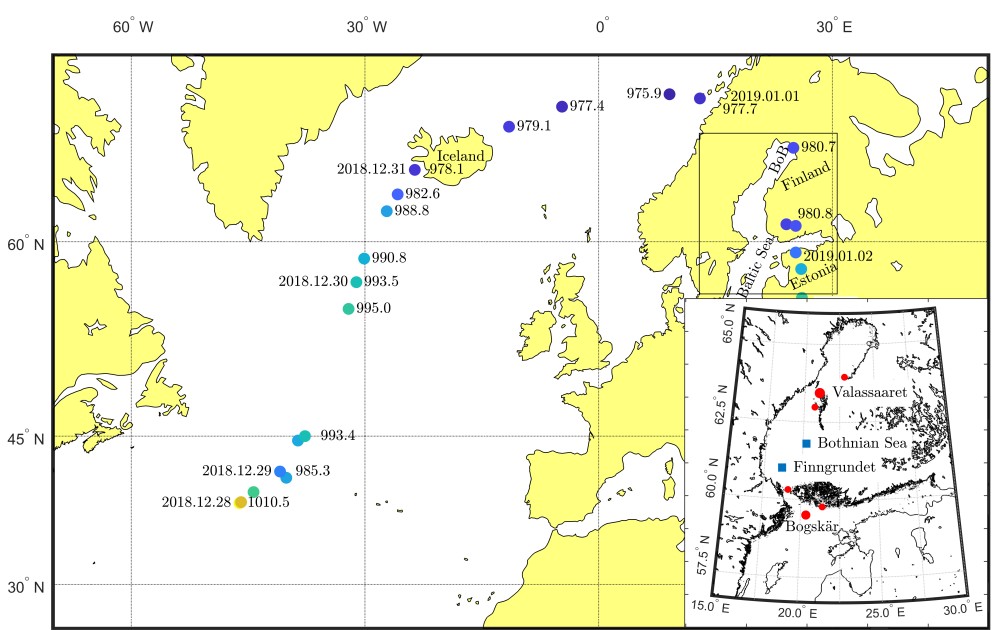

**Figure 1.** Storm track and sea level pressures (hPa) in the centre of the cyclone. The inset shows the Baltic Sea, the location of the wave buoys, and the two main weather stations. The four smaller red circles mark four additional weather stations (from north to south): Tankar, Stömmingbådan, Märket, and Utö.

## 2.2 Wave model data

Two existing Baltic Sea wave hindcasts complemented the Bothnian Sea wave buoy observations. The hindcast of Björkqvist et al. (2018) covered 1965–2005; the data originate from the wave model SWAN (Simulating WAves Nearshore, Booij et al., 1999) that was forced with the 6 h resolution BaltAn65+ wind product (Luhamaa et al., 2011). The hindcast of Tuomi et al.

5 (2019) covered 1978–2013 and originated from the wave model WAM (WAve Model, Komen et al., 1994) that was forced with winds from a downscaled ERA-Interim product having a 3 h resolution (Dee et al., 2011; Berg et al., 2013). Both hindcasts accounted for the seasonal ice cover and their data were available every hour, which we interpolated to 30 minute values.

Björkqvist et al. (2018) validated the SWAN data set in the Bothnian Sea against observations from three wave buoys. The bias of the significant wave height was −0.04 m (model underestimated) and the root-mean-square error (RMSE) was 0.30 m.

10 Tuomi et al. (2019) compared the WAM hindcast with altimeter data and determined a −0.21 m bias and 0.56 m RMSE for the





entire Baltic Sea. Although the WAM hindcast had an overall negative bias, the authors found that WAM slightly overestimated the highest wave heights (over 4 m). We used 2011–2013 Bothnian Sea wave measurements and found that the WAM hindcast of Tuomi et al. (2019) had a 0.03 m bias and a 0.36 m RMSE. Our comparison showed no overestimation of the highest wave heights. Comparing the two hindcasts (WAM-SWAN), the bias and RMSE were 0.03 m and 0.33 m for the coinciding ice-free
time of 1979–2005. In summary, both hindcasts were sufficiently accurate for this study.

For the duration of the storm only, we implemented a separate WAM wave hindcast forced with operational FMI-HARMONIE winds. The model set-up, including the used wind forcing, was similar to that of the wave forecast model in the Copernicus Marine Environment Monitoring Service (CMEMS) of the Baltic Monitoring and Forecasting Centres (BAL MFC) (see Tuomi et al., 2017; Vähä-Piikkiö et al., 2019).

**2.3   Meteorological data**

ECMWF's operational model provided global meteorological data every six hours, with a horizontal grid resolution that was aggregated to 0.1°. The track of the storm center between 28 December 2018, 00 UTC and 2 January 2019, 18 UTC was determined from the mean sea level pressure using a minimum locating algorithm in predefined areas.

As a forcing for the storm wave hindcast (see Sect. 2.2), we extracted wind data from the high-resolution (2.5 km) weather
prediction system HARMONIE (HIRLAM-B, 2020) that is used to force the CMEMS operational wave forecast.

We used two types of remotely sensed winds that were available around the time of the storm. The high resolution, near real-time, level-3 scatterometer wind product (Driesenaar et al., 2019) was downloaded from the CMEMS database, and the level-2 Sentinel-1 Synthetic Aperture Radar (SAR) Ocean product (Vincent et al., 2020) was extracted from the Copernicus Open Access Hub. The scatterometer wind speed is retrieved from level-2 wind vectors using the CMOD7 Geophysical Model
Function (Verhoef, 2018). The data are interpolated to a regular 12.5 km grid, and this method has shown a 0.04 m s$^{-1}$ bias when validated against global buoy data (Verhoef and Stoffelen, 2018). In the Sentinel-1 product the wind field is retrieved from a level-1 SAR image using the CMOD-IFR2 Geophysical Model Function with a priori wind direction information from ECMWF's atmospheric model (Mouche and Vincent, 2019). The wind speed is gridded to a 1 km resolution, and this product has shown an average bias of −0.4 m s$^{-1}$ when validated against buoy measurements and coastal stations around Ireland
(de Montera et al., 2019).

Wind measurements from FMI's weather stations north of the Bothnian Sea (Valassaaret) and the northern part of the Baltic Proper (Bogskär) were used to quantify the maximum wind speed during the storm. The data were 10 minute averages, and the measuring heights were 26 m and 31 m respectively. Data from four additional stations (Tankar, Stömmingbådan, Märket, and Utö) were used to validate the HARMONIE winds and the remote sense product in Sect. 4.1. For the locations of all stations,
see Fig. 1.




## 3 Methods

### 3.1 Accounting for sampling variability

Wave buoy observations inherently include sampling variability associated with a stochastic process and this makes them fundamentally different from modelled data. Normally the statistical variability adds only scatter, but it creates a bias for

extreme wave heights (Forristall et al., 1996). This bias can be though of as a survivorship bias in the more scattered data: the highest values stem from a positive statistical fluctuation, thus excluding negative variations from e.g. annual maxima. We accounted for the sampling variability with two alternative methods.

### 3.1.1 Method 1: Filtering measurements

The first approach was to simply remove the random variability with a low-pass filter, for which Forristall et al. (1996) sug-

gested a 3 h smoothing time. We applied a 1 h standard deviation Gaussian filter, and 2 h and 3 h moving averages to the squared time series ($H_s^2$). The Gaussian filtered observations resolved similar time scales as the model data, and this filter was therefore adopted for this study (Fig. 2). Nevertheless, the 3 h moving average had a similar performance.

Filtering the data reduced the yearly maxima, with the highest value dropping from 8.1 m to 7.0 m (Fig. 3 a and c). Nevertheless, the filtering process removed information from time scales longer than 3 h (Fig. 2), and this 14 % reduction

might very well be too large. It is unlikely that the wave conditions in a small basin are stationary for 3 hours, making the smoothing time scale a trade off between preserving the true wave information and creating a homogeneous data set when combined with model data.

### 3.1.2 Method 2: Adding variability to model data

The second approach preserved the original measurements but added random scatter to the model data by assuming that

synthetic samples of the wave field variance were $\chi^2$-distributed. In other words:

$$\frac{\hat{m}_0}{m_0} \sim \frac{\chi^2_{\text{dof}}}{\text{dof}}, \tag{1}$$

where $\hat{m}_0$ denotes the sampled variance and dof denotes the degrees of freedom. Following Donelan and Pierson (1983) we used the spectrum to calculate the degrees of freedom for the 8.1 m event (dof= 200), which was used throughout. This value was probably too small for low wave conditions, but still matched the real scatter of the wave measurements overall (Fig. 2).

Also, it was unnecessary to model the scatter of small wave heights precisely, since the distributions were only fitted to the highest values, such as annual maxima.

The artificial samples varied around the deterministic model values, leading to an increase in annual maxima (Fig. 3 b–c). Compared to the 7.2 m maximum in the original model data, the highest new maximum in a 100 realisation ensemble was 8.0 m (11 % increase), with the mean of the 100 new maxima being 7.6 m (6 % increase).


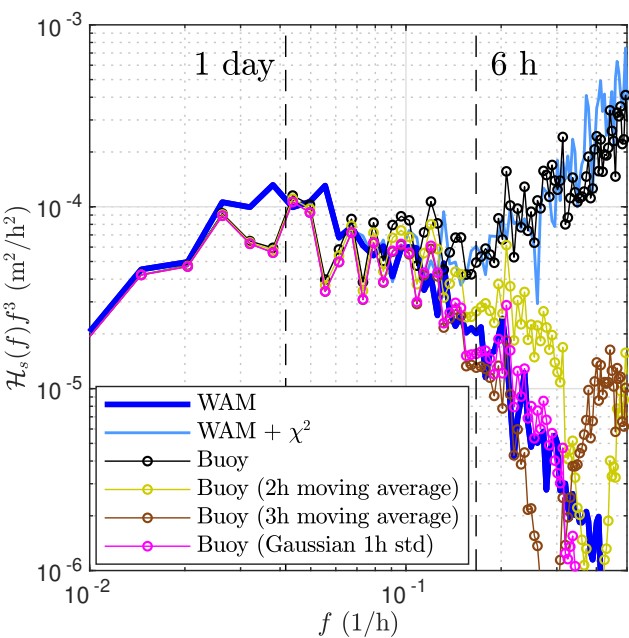

**Figure 2.** A comparison between model data with added $\chi^2_{200}$ sampling variability and observations that have been low-pass filtered by different methods. $\mathcal{H}_s(f)$ is the power spectrum of the significant wave height time series that has been multiplied by $f^3$ to accentuate shorter fluctuations. Data were from a coinciding ice-free period 28 May 2011 – 1 February 2012.

## 3.2 Combined measurement–model time series

We constructed 55-year time series at the Bothnian Sea wave buoy from the 2011–2019 wave buoy data, the 1965–2005 SWAN hindcast, and the 1979–2013 WAM hindcast. Model data were used for 1965–2010 by choosing one hindcast as the primary source of data and completing the time series using the other one. We then accounted for the difference in sampling variability 5 by either low-pass filtering the measurements (Sect. 3.1.1) or by adding synthetic $\chi^2$ sampling variability to the model data (Sect. 3.1.2).

The four resulting time series are denoted SWAN-$\chi^2$, SWAN-filtered, WAM-$\chi^2$, and WAM-filtered, according to the primary wave hindcast and the method used to account for the sampling variability (Table 1). For the $\chi^2$ data sets the results were an average of a 100 realisation ensemble.


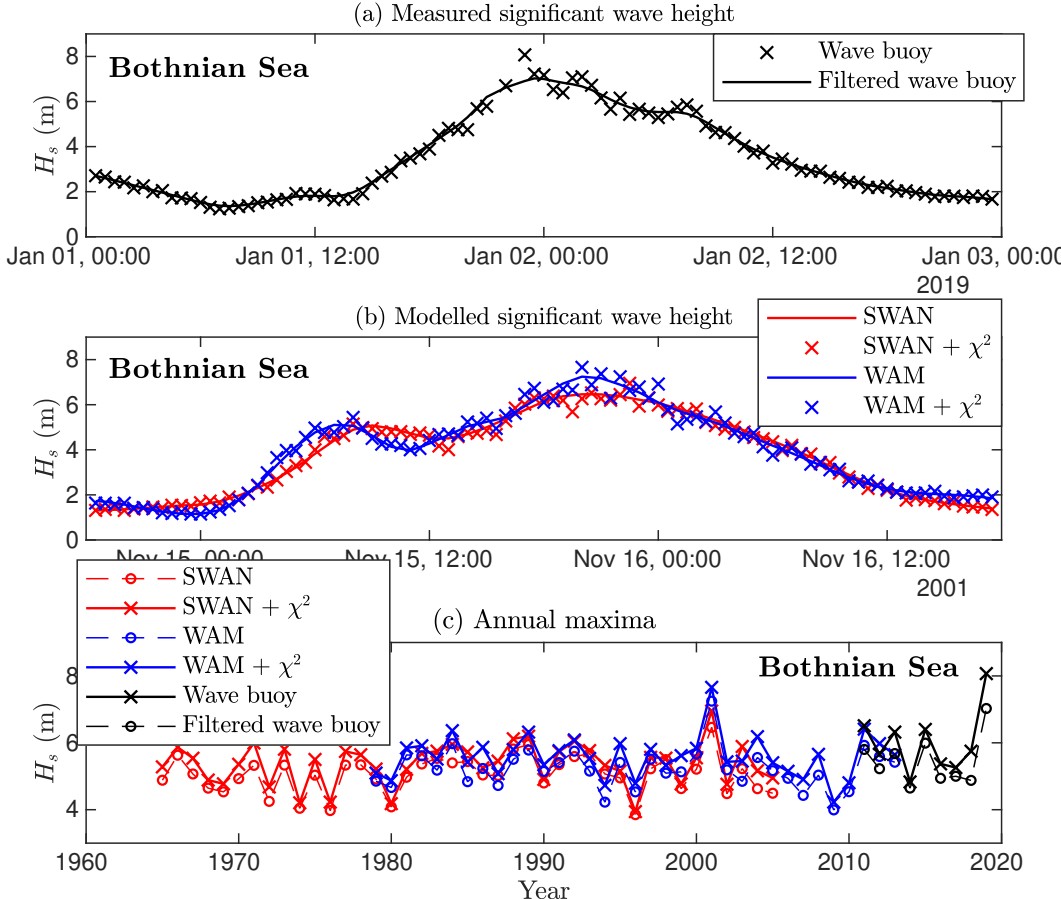

**Figure 3.** Short term variability removed from observations through low-pass filtering (a). Artificial $\chi^2$-distributed samples added to the wave model data from the most severe storm in the hindcasts (b). In both instances annual maxima increase when calculated from data with variability (c).

### 3.3 Fitting distributions and inferring return periods

Block maxima, $X$, can converge only to a member in the family of Generalised Extreme Value (GEV) distributions (e.g. Coles, 2001), having a CDF of:

$$
G(X) = \begin{cases} \exp\left\{-\left[1+\xi\left(\frac{X-\mu}{\sigma}\right)\right]^{-\frac{1}{\xi}}\right\} & , \xi \neq 0 \\ \exp\left\{-\exp\left[-\left(\frac{X-\mu}{\sigma}\right)\right]\right\} & , \xi = 0, \end{cases}
\tag{2}
$$

5  where $\mu$ is the location parameter, $\sigma$ is the scale parameter, and $\xi$ is the shape parameter. A negative value of $\xi$ means a thin-tailed distribution, and for $\xi = 0$ Eq. 2 equals the Gumbel distribution.





**Table 1.** The four different 55-year time series. First, either the SWAN or the WAM hindcast was used as the primary source of model data. Second, two alternative methods to account for the difference in sampling variability between observations and model data were applied (see Sect. 3.1). The data set with the added $\chi^2$ variability was a 100 realisation ensemble.

|  | $\chi^2$ variability added to model (observations as is) | Observations low-pass filtered (model data as is) |
|---|---|---|
| **SWAN (1965–2005)** | | |
| WAM (2006–2010) | **SWAN-$\chi^2$** | **SWAN-filtered** |
| *Buoy (2011–2019)* | | |
| SWAN (1965–1978) | | |
| **WAM (1979–2010)** | **WAM-$\chi^2$** | **WAM-filtered** |
| *Buoy (2011–2019)* | | |

The second set of distributions were fitted to peak-over-threshold (POT) data, where the original data, $x$, was converted to values exceeding a fixed threshold, $u$, i.e. $y = x - u$. The maximum values of $y$ were chosen and a minimum distance of 24 h between two values was imposed to insure independence. These data can converge only towards a Generalised Pareto Distribution (GPD) (e.g. Coles, 2001) with a CDF of:

$$H(y) = \begin{cases} 1 - \left[1 + \xi\left(\frac{y-\mu}{\sigma}\right)\right]^{-\frac{1}{\xi}} & , \xi \neq 0 \\ 1 - \exp\left(-\frac{y-\mu}{\sigma}\right) & , \xi = 0, \end{cases} \tag{3}$$

with the meaning of the parameters as for Eq. 2. For $\xi = 0$, $H(y)$ equals the exponential distribution.

We fitted these theoretical distributions to our data using the Maximum Likelihood Method. In the POT method we used the thresholds 4 m, 4.5 m, and 5 m, which correspond to the 99.5–99.9 percentiles in the combined data sets.

We inferred the return period by evaluating the CDF for the maximum storm wave height and calculating the return period (RP) in years as (e.g. Holthuijsen, 2007):

$$\text{RP} = \begin{cases} \dfrac{1}{1 - G(X)} & , \text{for annual maxima} \\ \dfrac{T_0}{1 - H(y)} & , \text{for POT data,} \end{cases} \tag{4}$$

where $T_0$ is the average time (in years) between two storms.





For the Gumbel and exponential fits the 95 % confidence intervals were determined from the confidence limits of the CDF. For the three-parameter distributions we used the 95 % confidence limits of the shape parameter $\xi$ (Table 2) and the best estimates of the other parameters.

## 4    Overview of the storm

### 4.1    Storm track and wind fields

On 28 December 2018 the storm started to develop over the Atlantic ocean close to 40° northern latitudes (Fig. 1). During the first couple of days the low pressure system deepened, then weakened, and finally split, with the northern part moving towards Iceland. Near Iceland the cyclone started to deepen again and moved northeast towards northern Scandinavia. After reaching the northern shore of Bay of Bothnia the storm center dove south and moved over Finland on 1 January 2019. When reaching Estonia the storm had already weakened, eventually filling up after 3 January.

On 1 January 2019, 21 UTC the weather station in the northern Bothnian Sea (Valassaaret, Fig. 1) measured a 29.1 m s$^{-1}$ wind speed from roughly 35°. Although the prevailing wind direction is south-south-west, the strongest winds are usually from northeast. The maximum in the entire Baltic Sea was observed 7 hours later at Bogskär, where northerly winds reached 32.5 m s$^{-1}$. Remotely sensed winds were not available from the peak of the storm, but larger parts of the Baltic Sea were covered in the evening before and the morning after the storm (Fig. 4).

At 19 UTC on 1 January winds north of the Bothnian Sea were blowing 22–24 m s$^{-1}$ (Fig. 4 a–b). The winds of the scatterometer and HARMONIE model products were almost identical, with both also agreeing with the coinciding measurements from the three northernmost weather stations. We surmise that the discrepancy at the Utö station (just south of the Archipelago Sea) was caused by the incomplete treatment of islands in the level-3 processing of the scatterometer product. Nonetheless, this error seems local, as both products match the measured wind speed from Bogskär, located slightly west-southwest of Utö.

At 05 UTC next morning the winds with a maximum intensity over 24 m s$^{-1}$ had moved past the Bothnian Sea to the northern Baltic Proper (Fig. 4 c–d). Although both SAR and HARMONIE show similar large-scale patterns, the wind speeds from SAR are systematically lower. This is probably indicative of too weak SAR winds, since overall HARMONIE agrees better with the coinciding weather station measurements. SAR's slight underestimation of higher wind speeds in the Baltic Sea is in line with a previous validation by Rikka et al. (2018).

In summary, the modelled HARMONIE winds were in good accord with both the remote sensing products and weather station data. The comparison also confirmed what was suggested by the in situ measurements, namely that the cyclone generated strong winds along the entire Bothnian Sea when travelling south.

### 4.2    Waves during the storm

Prior to 2019 the highest significant wave height measured in the Bothian Sea was 6.5 m (in 26 December 2011). During the storm this value was exceeded for 3.5 hours with the significant wave height reaching a maximum of 8.1 m on 1 January
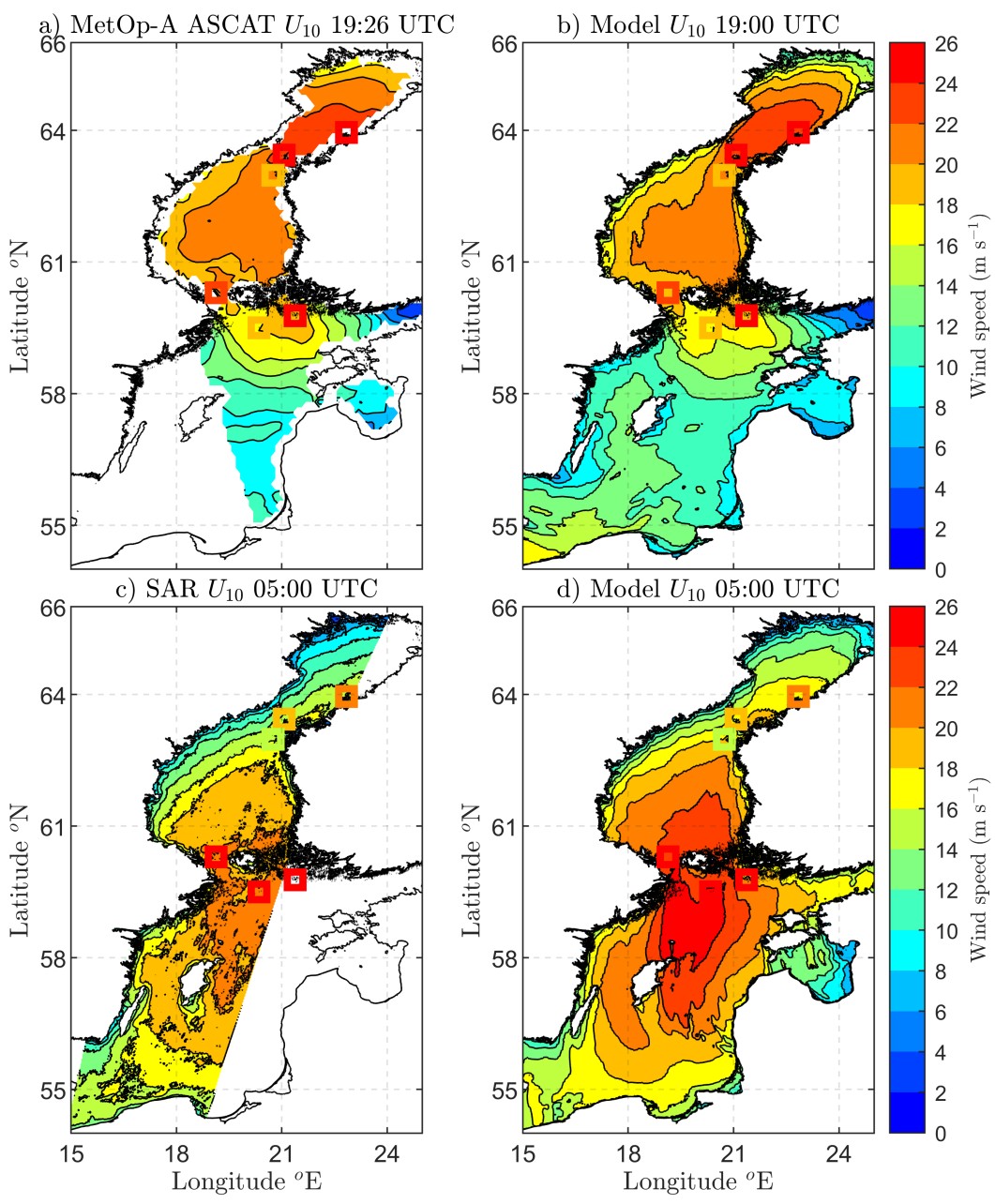

**Figure 4.** Wind speed $U_{10}$ retrieved from a scatterometer and SAR (a and c, left) and modelled by HARMONIE (b and d, right) before and after the storm. The overlaid squares correspond to in-situ measurements (from north to south): Tankar, Valassaaret, Strömmingsbäadan, Märket, Utö, and Bogskär.


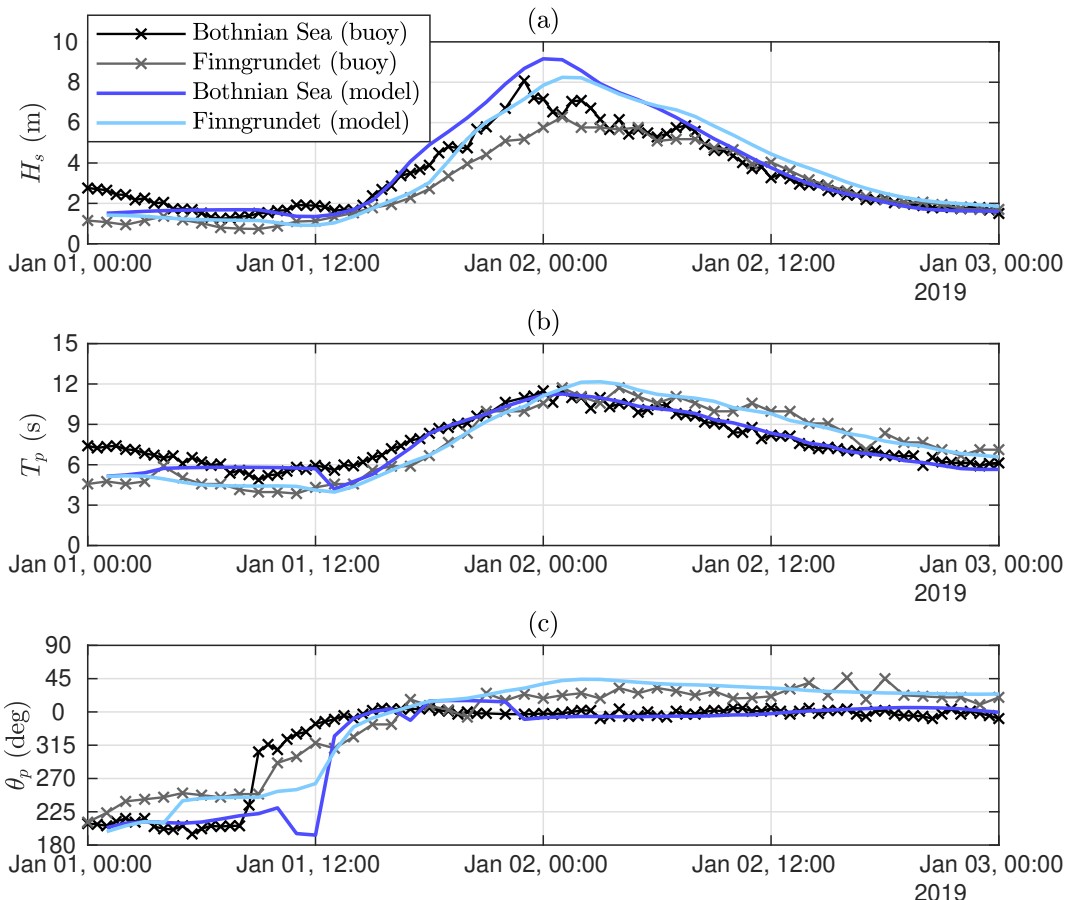

**Figure 5.** The significant wave height ($H_s$), peak period ($T_p$) and mean direction at the spectral peak ($\theta_p$) at the Bothnian Sea and Finngrundet wave buoys. The WAM model hindcast is forced for HARMONIE with a similar set-up as the CMEMS operational forecast.

23:00 UTC (Fig. 5, a). This maximum wave height is equal to those measured during storms in the Baltic Proper main basin (Björkqvist et al., 2017). The highest significant wave height at the Finngrundet wave buoy in the southern part of the basin was 6.3 m. The waves in the southern part were lower during the growth phase, but both wave buoys measured similar wave heights during the decay.

5    The 12 s maximum peak period at Finngrundet equalled that of the Bothnian Sea wave buoy during the height of the storm (Fig. 5 b). Nevertheless, during the relaxation of the storm the waves at Finngrundet were up to 2 s longer compared to the Bothnian Sea buoy, which is explained by the difference in fetch. The wave direction at the Bothnian Sea wave buoy was steady from the north, while being from north-north-east at Finngrundet (Fig. 5, c). This 20–30° difference was caused by depth-induced wave refraction or the slanting fetch geometry in the basin (Holthuijsen, 1983).

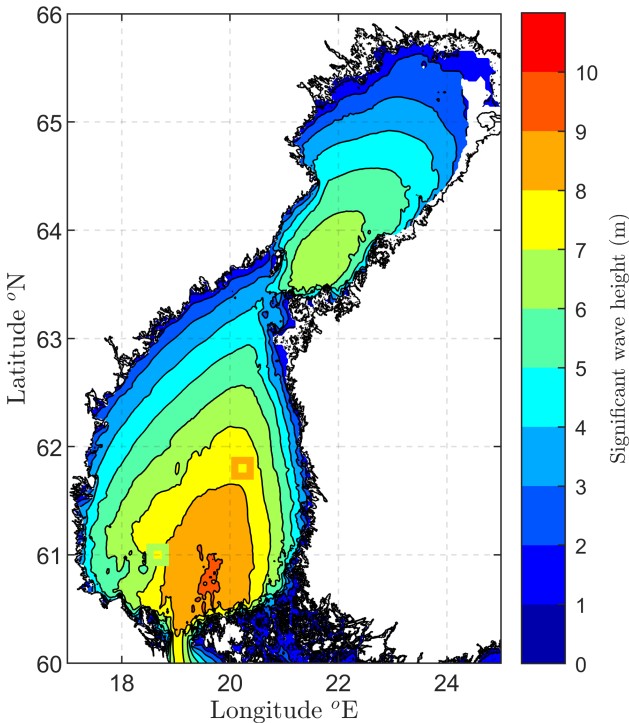

**Figure 6.** The maximum significant wave height for the January 2019 storm from the calibrated wave hindcast. The maximum measured significant wave heights at the Bothnian Sea and Finngrundet wave buoys are shown by overlaid squares.

Basin-wide wave fields from the storm were available from the HARMONIE-forced hindcast, which – while capturing the period and direction accurately – overestimated the significant wave height at both wave buoy locations (Fig. 5). A calibration against Bothnian Sea and Finngrundet buoy observations from 1–3 January determined a 20 % positive model bias in the significant wave height, which was corrected for in the entire Gulf of Bothnia model data. Fig. 6 shows that the significant wave height exceeded 8 m in a large part of the Bothnian Sea, even reaching 9 m in the southernmost part of the basin. We consider these calibrated model results reliable, as they even capture the small scale features near the Finngrundet buoy accurately. In the Bay of Bothnia (north of the Bothnian Sea, Fig. 1) the significant wave height exceeded 6 m, which is higher than the maximum value of 4.6 m that FMI's wave buoy has measured in the middle part of the basin since 2012. Nevertheless, no wave buoy data were available from the Bay of Bothnia for the storm, and these modelled values are therefore less reliable.

Even though the coast of Bay of Bothnia – including the narrow passage to the Bothnian Sea – typically freeze already in December (SMHI and FIMR, 1982), the sub-basin was almost ice-free on 1 January 2019. We still estimated that the waves propagating from the Bay of Bothnia probably had none to little effect on the sea state at the Bothnian Sea wave buoy (see Appendix A). On the other hand, the coasts of the Bothnian Sea are also frequently frozen in January. If this would have been the case in 2019, the narrower fetch geometry would have lowered the maximum wave height.





**Table 2.** The shape parameters, $\xi$, of GEV and GPD fits to annual maxima an POT data, where GPD (4.0) denotes a $u = 4.0$ m threshold for the POT. A negative value of $\xi$ means a thin-tailed distribution. For a description of the data sets (e.g. WAM-$\chi^2$), see Table 1.

|  | SWAN-$\chi^2$ | | SWAN-filtered | |
| --- | --- | --- | --- | --- |
|  | $\xi$ | 95 % conf. | $\xi$ | 95 % conf. |
| GEV | $-0.11$ | $(-0.24, 0.02)$ | $-0.16$ | $(-0.30, -0.03)$ |
| GPD (4.0) | $-0.12$ | $(-0.20, -0.04)$ | $-0.14$ | $(-0.24, 0.04)$ |
| GPD (4.5) | $-0.07$ | $(-0.20, 0.05)$ | $-0.10$ | $(-0.27, 0.07)$ |
| GPD (5.0) | $-0.02$ | $(-0.23, 0.19)$ | $-0.07$ | $(-0.39, 0.25)$ |
|  | WAM-$\chi^2$ | | WAM-filtered | |
|  | $\xi$ | 95 % conf. | $\xi$ | 95 % conf. |
| GEV | $-0.08$ | $(-0.22, 0.05)$ | $-0.11$ | $(-0.25, 0.02)$ |
| GPD (4.0) | $-0.12$ | $(-0.19, -0.05)$ | $-0.11$ | $(-0.20, -0.01)$ |
| GPD (4.5) | $-0.08$ | $(-0.19, 0.03)$ | $-0.03$ | $(-0.19, 0.13)$ |
| GPD (5.0) | $0.02$ | $(-0.19, 0.22)$ | $0.02$ | $(-0.28, 0.33)$ |

## 5 Estimating the return period

We now present results obtained from fitting the theoretical distributions of Sect. 3.3 to the data sets presented in Table 1. On average the GEV and GPD distribution were thin-tailed ($\xi < 0$), with the WAM data sets resulting in slightly higher values for the shape parameter $\xi$ than the SWAN data (Table 2). Nonetheless, the 95 % confidence intervals typically contained $\xi = 0$.

The shape parameter of the GDP fit increased with higher thresholds, which indicated that the highest wave heights were better captured by an exponential type distribution ($\xi \approx 0$). For the block maxima, the GEV shape parameter typically resembled that of a GPD fit using the lowest threshold $u = 4.0$ m (with the exception of WAM-$\chi^2$).

Table 3 summarises the return periods estimated from the different distributions and data sets. We make four observations:

**I) Fixing $\xi = 0$ vs. GEV/GPD**: The negative shape parameters of the general distributions (GEV/GPD) resulted in longer return period estimates compared to the specific distributions (Gumbel/exponential). Increasing the threshold in the POT method resulted in shorter return periods when using the GPD distributions. In contrast, the exponential fits gave longer return periods when the threshold was increased. For the highest threshold, $u = 5.0$ m, the results of the GPD's were close to that of the exponential distributions, which is consistent with $\xi$ being near 0.

**II) POT vs. Annual maxima**: The results from POT data were most consistent with those from annual maxima when using a threshold of 4.5 m (for $\chi^2$ data) or 4.0 m (for filtered data). This difference might be explained by the lower values in the filtered data, even though 4.5 m is in the 99.8 percentile for both data sets.





**III) Filtered vs. $\chi^2$ data**: Using filtered data lead to consistently shorter estimates for the return period compared to using the $\chi^2$ data sets. This was the case for all distributions and both sources of model data (SWAN/WAM). When the specific distributions were fit to filtered data, they gave return periods that were, on average, 64 years (44 %) shorter than for $\chi^2$ data. For the general distributions this difference was 309 years (57 %).

**IV) SWAN vs. WAM hindcast**: The general distributions were sensitive to the choice of primary model hindcast: a change from SWAN to WAM reduced the estimates from the GEV/GPD's by an average of 149 years (40 %). In contrast, the Gumbel and exponential distribution were consistent between the different sources of model data, with a difference of only 6 years (7 %).

The GEV and GPD fits were heavily influenced by significant wave heights below roughly 5 m, thus representing the extremes less accurately than the Gumbel or exponential distributions (Fig. 7). This influence – seen as a downward curvature in the fits of Fig. 7 – was more severe for the GEV compared to the GPD ($u = 4.5$ m). Also, fitting a GPD with a 4.0 m threshold exacerbated the effect (not shown here, but figure available as supplementary material; also see Table 3).

    In summary, the general distributions gave poor fits for the highest values, were sensitive to the choice of model data, 15 and gave up to infinite return periods with the lower bound $\xi$-values (Table 3). These findings motivated us to discard the GEV/GPD distributions. Fixing $\xi = 0$ was reasonable in the general frame work of extreme value theory, since this value was typically included in the confidence intervals of the parameter (Table 2). Restricting ourselves to the Gumbel and exponential distributions, the filtered data gave an average estimated return period of 72 years, with a 95 % confidence of (28, 224). The similar value using $\chi^2$ data was 136 years (50, 423), and the grand average using both types of data was 104 years (39, 323).

Visually, the exponential distributions were good fits for the entire range of significant wave heights when using $u = 4.5$ m (Fig. 7). The annual maxima showed an "inverse S-shape", which was incompatible with all the theoretical distributions. One reason for the better fit of the exponential distribution compared to the Gumbel distribution was probably the wastefulness of block maxima: there are only 55 annual maxima, but 106 POT data points (SWAN-filtered, $u = 4.5$ m). All in all, we deemed the exponential fit with a threshold of $u = 4.5$ m as the most reliable. Yet, the return period from this distribution – when averaged 25 using all data sets – was 104 years (44, 268), which is practically identical to 104 (39, 323) determined from the larger average (i.e. only fixing $\xi = 0$ and nothing else). Picking this exact distribution would therefore have been of little consequence.

## 6   Discussion

### 6.1   Interpreting the results

The concept of a return period can be misleading. Correctly interpreted a 100 year return period equals a 1 % annual exceedance 30 probability. Thus, the probability that an event with a 104 year (39, 323) return period actually occurs during the next century is 62 % (27 %, 93 %) – assuming a constant climate. As widely recognised, confidence intervals are an insufficient measure of uncertainty, especially since combining measurements and model data has large possible sources of error. We will now discuss our estimates, and the related uncertainties, for the present and future climate.





**Table 3.** The estimated return period of the maximum significant wave height at the Bothnian Sea wave buoy during the storm using block maxima and POT data. The notation GPD (4.0) means a $u = 4.0$ m threshold for the POT data. For a description of the data sets (e.g. WAM-$\chi^2$), see Table 1.

| | SWAN-$\chi^2$ | | SWAN-filtered | |
|---|---|---|---|---|
| | Return period (y) | 95 % conf. (y) | Return period (y) | 95 % conf. (y) |
| Gumbel | 116 | $(41, 332)$ | 55 | $(22, 136)$ |
| Exponential (4.0) | 69 | $(37, 137)$ | 44 | $(23, 88)$ |
| Exponential (4.5) | 136 | $(56, 353)$ | 82 | $(33, 224)$ |
| Exponential (5.0) | 235 | $(68, 931)$ | 117 | $(33, 518)$ |
| GEV | 647 | $(86, \infty)$ | 419 | $(60, \infty)$ |
| GPD (4.0) | 904 | $(72, \infty)$ | 349 | $(36, \infty)$ |
| GPD (4.5) | 485 | $(47, \infty)$ | 232 | $(28, \infty)$ |
| GPD (5.0) | 305 | $(35, \infty)$ | 184 | $(24, \infty)$ |
| | WAM-$\chi^2$ | | WAM-filtered | |
| | Return period (y) | 95 % conf. (y) | Return period (y) | 95 % conf. (y) |
| Gumbel | 120 | $(42, 349)$ | 52 | $(21, 126)$ |
| Exponential (4.0) | 58 | $(32, 110)$ | 40 | $(22, 77)$ |
| Exponential (4.5) | 114 | $(50, 277)$ | 84 | $(35, 218)$ |
| Exponential (5.0) | 239 | $(73, 895)$ | 101 | $(31, 402)$ |
| GEV | 378 | $(66, \infty)$ | 147 | $(39, 79\,349)$ |
| GPD (4.0) | 814 | $(75, \infty)$ | 159 | $(28, 750\,836)$ |
| GPD (4.5) | 426 | $(48, \infty)$ | 112 | $(22, 7\,666\,949)$ |
| GPD (5.0) | 204 | $(30, \infty)$ | 90 | $(20, \infty)$ |

## 6.2 Combining the data sets

The ideal low-pass filter doesn't exist. Therefore, a 3 h smoothing time also removes longer variations from the measurements (Fig. 2), and these variations are most likely physically relevant. It seems like it would be better to preserve the original observations and add $\chi^2$ variability to the model data. But if the model data match the (too aggressively) filtered buoy data, it

5 cannot be made fully compatible with the original observations by simply adding random scatter. A gentler filter preserves the properties of the measured wave field better, but at the expense of a homogeneous measurement–model data set.

By possibly filtering out parts of what made the event unique, the estimated return periods may consequently be biased low. In the WAM-filtered data set the maximum significant wave height was actually 7.2 m, which was modelled by WAM on 15


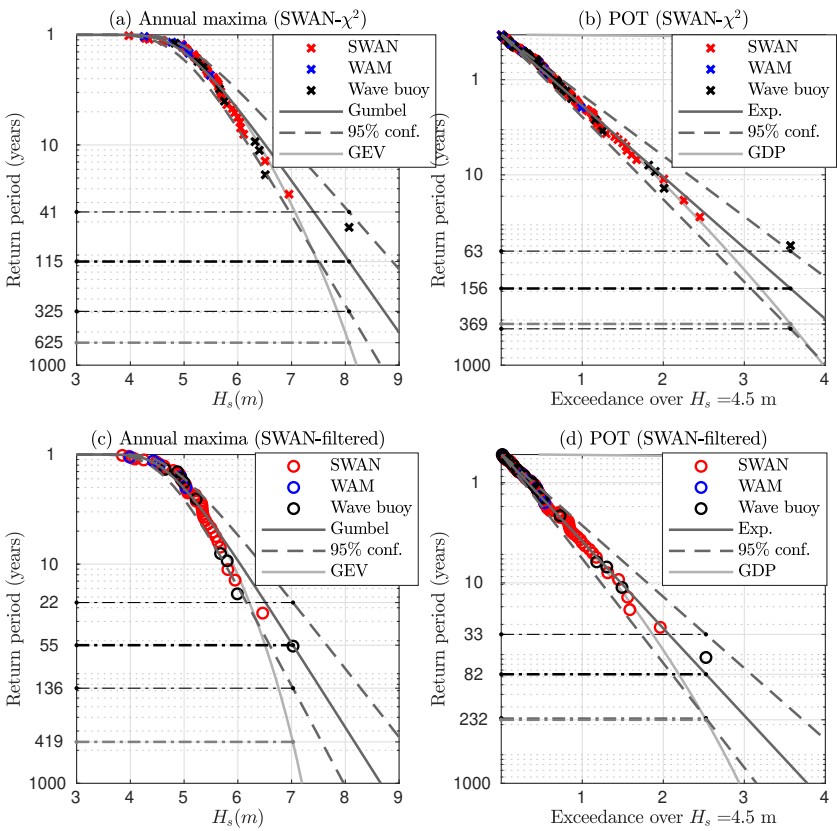

**Figure 7.** Fits to annual maxima and POT values. Top panels: original observations and model values with an added synthetic $\chi^2$-variability. Bottom panels: original model data and low-pass filtered observations. A single realisation of added $\chi^2$-variability being close to averages of Table 3 has been chosen. Illustrations of other data combinations and thresholds are available as supplementary material.

November 2001 (Fig. 3) – the filtered measured maximum from 2019 was only 7.0 m. On the other hand, the return periods from the $\chi^2$ data may be too long; the short term variations in the model data are inadequate even after the added scatter, thus making all observed events seem rarer than they actually are.

The different time resolution in the wind forcing between the SWAN hindcast (6 h) and the WAM hindcast (3 h) possibly explain some of the differences between their results. The findings of Björkqvist (2020) suggest that a 1 h wind forcing can capture most of the variability in the wave field (aside from random scatter). Studies using high temporal-resolution wind products are therefore needed to further study how to optimally combine wave observations and model data.

## 6.3 In the present climate

The cyclone did not follow a typical path that induce south-westerly winds. Instead, it generated strong winds along the longer north–south axis of the Bothnian Sea by approaching the Baltic Sea from the north. While the path isn't unique, there are





indications that the storm was rare. First, the Bogskär weather station recorded a $32.5 \text{ m s}^{-1}$ 10 minute average wind speed – the highest value ever measured by FMI in the Baltic Sea. Second, new sea level minima were recorded at all four tide gauges along the Finnish coast of the Bothnian Sea. These stations have been active for almost a century (since 1922, 1925, 1926, and 1933). Therefore, a return period much shorter than our lower bound estimate of roughly 40 years seems unlikely.

Extreme value theory states that if block maxima converge, then their distribution will be in the GEV family. But yearly blocks are not necessarily long enough for this to happen. Our highest annual maxima were separated from the rest of the data (Fig. 7), possibly because the assumption of drawing from an identical distribution was violated. Then again, the stability of these highest values are infamously poor. Be that as it may, our extremes were modelled better with the shape parameter fixed to $\xi = 0$ (Gumbel/exponential) than with the general fits (GEV/GPD). We considered the results from the general fits to

be unrealistic and back our opinion with a qualitative argument: a half-a-millennium return period is unlikely when the event was observed within nine years after measurements begun. Also, the 95 % confidence values of the shape parameter, such as $\xi = -0.24$, modelled an infinite return period for a documented event.

     Haakenstad et al. (2020) estimated 7–8 m 100 year significant wave heights near the Bothnian Sea wave buoy with a Gumbel distribution (their Fig. 13 d), and Aarnes et al. (2012) found similar values using a GEV fit. These studies focused

on the northeast Atlantic, thus having a coarse (11 km) model resolution compared to Baltic Sea specific simulations (e.g. Björkqvist et al., 2018). Their results still agree with our Gumbel distribution's 100 year value (roughly 7.5–8.0 m depending on the data set; Fig. 7 a and c).

     At first glance our GEV fit (419/647 year return period for a 7.0/8.1 m event; Fig. 7 a and c) contradicts the 100 year estimate of Aarnes et al. (2012). Still, the 100 year significant wave height estimated from our GEV fit was roughly 6.7–7.5 m – again,

depending on the data – thus agreeing reasonably well with both Haakenstad et al. (2020) and Aarnes et al. (2012). Estimates of the return period of a rare event are more sensitive to the distribution tail compared to that of return levels (e.g. 100 year wave heights). On the other hand, this means that a thorough return period analysis of a well documented event can give valuable information about how plausible different tails of distributions are.

### 6.4    In a future climate

No long term historical trend in Baltic Sea wave heights has been reliably documented. Rather, different studies using disparate data of variable quality have gotten conflicting results (e.g. Broman et al., 2006; Zaitseva-Pärnaste et al., 2009; Räämet et al., 2010). Groll et al. (2017) did project an increase in median significant wave heights – but not extreme values – for 1961–2100. The lack of a trend in annual maxima is consistent with our Bothnian Sea data, but only if observation uncertainty is accounted for properly; ignoring the sampling variability creates a spurious $1.5 \text{ cm y}^{-1}$ trend – an 80 cm increase over the 55 year time

series. This trend vanishes if the data are treated with either method outlined in this paper.

     A pseudo-climate study by Mäll et al. (2020) reported an expanding area of extreme waves for past Baltic Sea cyclones under the RCP8.5 emission scenario. While the results were not conclusive, and included no north-to-south cyclones, it's possible that the 2019 storm could respond in a similar way. On a local scale, the strongest (predominantly north-easterly) winds in the northern Bothnian Sea have been projected to shift anti-clockwise (Ruosteenoja et al., 2019). On a broader scale, the shrinking





Arctic ice cover (Boé et al., 2009; David et al., 2020) leads to a higher latent heat transport from an open and warmer ocean. Still, we can only speculate how the likelihood or strength of storms resembling that of January 2019 could be affected by meteorological changes.

What is more certain is that the seasonal ice cover of the Baltic Sea is declining (e.g. Vihma and Haapala, 2009). In the
present climate the mean annual maximum ice extent covers the entire Bothnian Sea (SMHI and FIMR, 1982; Höglund et al., 2017). The ice-free Bothnian Sea on 1 January 2019 corresponded to average conditions projected under RCP8.5, but a retreat is projected also under the more moderate RCP4.5 scenario (Höglund et al., 2017). A declining ice cover will lead to longer fetches appearing more often. The maximum wave height in January 2019 was adequately model using only the wind speed and the Bothnian Sea fetch (Appendix A), and growth-curves could therefore be used to quantify the response of the highest
waves in the Bothnian Sea to its declining seasonal ice cover.

## 7   Conclusions

We investigated methods to combine wave measurements (containing sampling variability) and wave model results (without sampling variability) to a coherent time series. This study was motivated by the need to compensate for insufficient measurement data in order to more reliably estimate the return period of a wave event. The wave event in question caught our interest
because it exceeded previously measured and simulated values in the Bothnian Sea sub-basin of the Baltic Sea. During the storm the highest wind speed of $32.5 \text{ m s}^{-1}$ was measured in the Baltic Proper main basin, and the maximum significant wave height of $8.1 \text{ m}$ was measured in the Bothnian Sea. Nonetheless, a hindcast specifically calibrated for the storm indicated that the significant wave height reached $9 \text{ m}$ in the southern Bothnian Sea (Fig. 6).

To estimate the return period of the event, we covered the time 1965–2019 with nine years of wave measurements and two long-term wave model hindcasts, and combined them to two types of data:

1. without sampling variability (observations low-pass filtered, model values as is),

2. with sampling variability (synthetic variability added to model, observations as is).

The GEV and GPD distributions modelled unrealistically long return periods and were sensitive to the choice of wave
hindcast product. The highest significant wave heights were best captured with a shape parameter close to $\xi = 0$, and we therefore used the Gumbel and exponential distributions to estimate the return period of the storm wave event.

Both methods to account for the difference in sampling variability had their strengths and weaknesses, and it's still uncertain how to best combine modelled and measured wave height data. Averaging the results from both methods, we estimated that the return period was 104 years (95 % conf. 39–323 years). The estimate was 32 years longer (or shorter) if we only used data
with (or without) sampling variability.




Although the impact of meteorological changes in the Baltic Sea are uncertain, we surmise that the declining seasonal ice cover will result in wave events of this magnitude becoming more frequent. Additional studies are required to confirm this assertion and to quantify the effect of the retreating seasonal ice.

In conclusion, we found that properly accounting for the lack of sampling variability in model data is crucial if they are used for extreme value analysis in combination with in situ measurements. The different nature of modelled and measured wave data is universal and should, in principle, be taken into account in all studies where they are combined. Nevertheless, the uncertainty of the combination process remains large, further highlighting the value of long, homogeneous, observational time series. Future studies on this topic should use waves that have been modelled using wind data with a high time-resolution, preferably no longer than 1 h.

*Code and data availability.* The wave time series for the Bothnian Sea wave buoy location can be accessed through the DOI: 10.5281/zenodo.3878948. The Finngrundet wave measurements are available from the SMHI open data portal (https://opendata-download-ocobs.smhi.se/explore/). The remotely sensed data are available at Copernicus CMEMS data portal (https://marine.copernicus.eu/) and the Copernicus Open Access Hub (https://scihub.copernicus.eu/). The distributions were fitted using the Machine learning and Statistical toolbox in MATLAB R2018b. The $\chi^2$-simulations were performed using the Machine learning and Statistical toolbox in MATLAB R2018b. The random number generator was set to "default" to ensure reproducability.

## Appendix A: Role of waves propagating from the Bay of Bothnia

### A1   Hypothetical ice mask in the wave model

We implemented the HARMONIE forced wave model with an hypothetical ice mask covering the entire Bay of Bothnian and compared the results to the original hindcast. The modelled significant wave height and peak period at the Bothnian Sea wave buoy was identical between the two model runs, suggesting that the waves propagating from the Bay of Bothnia to the Bothnian Sea play a negligible part in the Bothnian Sea south of the wave buoy.

### A2   Fetch limited wave growth in the Bothnian Sea

The wind speed at a Valassaaret weather station between the Bothnian Sea and the Bay of Bothnia (Fig. 1) during the peak of the storm ($U > 25\,\mathrm{m\,s^{-1}}$) was $26.8\,\mathrm{m\,s^{-1}}$. A wind speed of $U = 27\,\mathrm{m\,s^{-1}}$ is compatible with the measured wave spectrum at the height of the storm (Fig. A1) when considering the equilibrium constant and the power-law transition frequency (Phillips, 1958; Kahma, 1981; Kahma and Calkoen, 1992). The directional spreading at the spectral peak was 16° and we therefore determined the 194 km northerly fetch as an average over a 30° sector.

A $27\,\mathrm{m\,s^{-1}}$ wind speed of would result in a peak period of 11 s (composite data set; Kahma and Calkoen, 1992). This is in reasonable agreement with the measured value of 12 s. Assuming the aforementioned wind speed and fetch, the locally


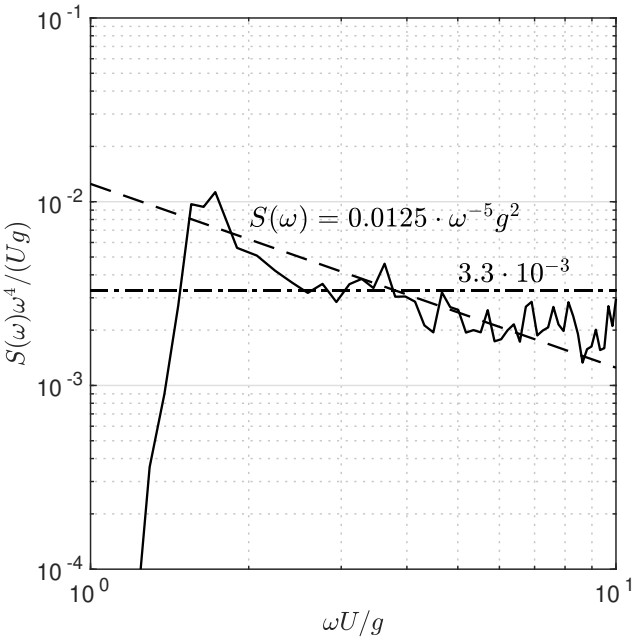

**Figure A1.** The wave spectrum measured with the Bothnian Sea wave buoy at the height of the storm (1 Jan 2019 23:00 UTC). The spectrum has been scaled assuming a wind speed of $U = 27$ m s$^{-1}$. The equilibrium level (dashed-dotted) and the saturation level (dashed) are also shown.

generated significant wave height in the Bothnian Sea would have been 7.4 m. Although this is less than the measured value of 8.1 m, it's a fair match to the low-pass filtered maximum of 7.0 m.

*Author contributions.* The paper was initiated by JVB and VA. AM was responsible for the analysis of the meteorological conditions. The in-situ wave measurements were analysed by JVB and HP, and remote sensing data were analysed by SR. The additional wave model simulations were performed by LT, and they were reviewed by SR and VA. Statistical analysis was performed by JVB. The manuscript was prepared by JVB, SR, VA, and AM, with contributions from all co-authors.

*Competing interests.* The authors declare that they have no conflict of interest

*Acknowledgements.* This work was partially supported by the Strategic Research Council at the Academy of Finland, project SmartSea (grant number 292 985), by the State Nuclear Waste Management Fund of in Finland (VYR) through the Finnish Research Programme on Nuclear Power Plant Safety 2019–2022 (SAFIR2022), by the Estonian Research Council (grant number PSG22), and by Personal Research Fund-



ing of the Estonian Ministry of Education and Research (grant number PUT1378). This study has utilised research infrastructure facilities provided by FINMARI (Finnish Marine Research Infrastructure network), E.U. Copernicus Marine Service Information (products BALTIC-SEA_ANALYSIS_FORECAST_PHY_003_006 & WIND_GLO_WIND_L3_NRT_OBSERVATIONS_012_002), and data provided by the Copernicus Open Access Hub. We acknowledge the effort of Dr. Jani Särkkä to post-process and provide the data of the WAM hindcast.



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
