# Peer review of "Wave height return periods from combined measurement–model data: A Baltic Sea case study"

_Natural Hazards and Earth System Sciences, 2020_

## Referee Comment (RC1) · Anonymous Referee #1 · 2 Jul 2020

This is an interesting and very well written paper. I have no objection to it being published as submitted, but I would very much like to see the authors' responses to my comments.

The most interesting and novel part of the paper is the treatment of sampling variability in extreme value estimation. Figure 2 is very informative. I have not seen a spectral analysis of Hs history before. It is interesting that filtering removed information from time scales longer than three hours, despite the fact that the Gaussian filter half power point was one hour. The moving averages of course had side lobes. It might be worthwhile to investigate more sophisticated digital filters. But the more interesting

question is what time scales are really represented in hindcast data that is reported every hour but is based on three or six hour wind fields. Perhaps more work with filtering the continuous measurement time series would help answer that question.

Different users are interested in different time scales. Ship designers often want to know the three hour sea state for use in model basins. Calculating extreme values of individual wave heights from shorter averaging times where Hs is not varying may be more accurate. Would calculations of individual wave heights from the hindcast data match those from thirty minute measurements (or chi-squared augmented hindcasts)?

The difference in return periods between the chi-squared and filtered analyses deserves comment. If the chi-squared augmentation worked perfectly, wouldn't they be equal? Looking at Figure 3, it seems that the extreme wave height in the measurements has a larger deviation from the smooth curve than most of the artificial chi-squared data. That makes me think that the measurement is an outlier to the chi-squared distribution. Why don't you plot the variability of the measurements against a chi-squared distribution to check that?

And finally, why do you think the hindcast of the recent storm was so bad?

---

## Referee Comment (RC2) · Anonymous Referee #2 · 30 Jul 2020

The manuscript that the authors presented is overall well written and discuss an interesting and imported issue of how to compare and combine model and observational data. Further the authors use this new combined dataset to investigate return values of extreme significant wave heights in the northern Baltic Sea. I generally recommend this manuscript for publication after minor revision, I would like to encourage the authors to address the following comments.

Comments on the homogeneity of the new combined datasets.

The authors stated that they construct a time series of two hindcast simulation and one observational dataset. To account for the different variability in the datasets they us-

ing a low pass filter for the observation and adding variability to the model data. They analyse and discuss the different methods and the results. I would suggest that combining datasets from different hindcasts, with different atmospheric forcing and different temporal resolution should lead to some inhomogeneity. The authors stated that there is some kind of trade-off between homogeneity and preserving the "true" wave field when merging observations and model data. The manuscript would benefit to also discuss possible effects of combining different models datasets to the analysis of extreme values.

If correctly understand, the two data series marked with WAM and SWAN only differ between the period 1979-2010 (30 out of 55 years). On one hand, the results of the return period differ quite a bit (especially using the GEV), on the other hand, some results seem to have almost no difference. Is there any explanation of these different outcomes? Could it be the one model dominates the distribution?

Technical comments

There is a discrepancy between the time series plotted in Figure 5, the text in section 4.2 and the colour coding in Figure 6. Whereas it is described and visible in Figure 5 that the model overestimates the observations, the coloured squares in Figure 6 indicate higher wave height for the observations (orange (8-9m) square over yellow (7-8m) area). Also, it is stated that large parts of the Bothnian Sea show a significant wave height above 8m, but the orange to red area in Figure 6 is about a third to a quarter of the Bothnian Sea area. On the other hand, it is stated that the maximum in the Bothnian Bay is above 6m, which is confirmed by the image.

The titles of the four panels in Figure 7 show two times "SWAN-X2" and two times "SWAN-filtered". If this is correct it is at least confusing for the reader as I would expect filtered vs X2 and annual maxima vs POT. Also it is not mentioned which buoy is represented in the Figure 7, Bothnian Sea buoy or Finngrundet buoy.

Finally, probably ":5(216)" are missing at the end of the DOI for reference Forristall et

al. 1994, DOI https://doi.org/10.1061/(ASCE)0733-950X(1996)122:5(216)
* * *

---

## Author Response (AR1)

R1: This is an interesting and very well written paper. I have no objection to it being published as submitted, but I would very much like to see the authors' responses to my comments.

Our response: Thank you. We appreciate you agreeing to review our paper. Please see our responses to your comments below.

R1: The most interesting and novel part of the paper is the treatment of sampling variability in extreme value estimation. Figure 2 is very informative. I have not seen a

spectral analysis of Hs history before. It is interesting that filtering removed information from time scales longer than three hours, despite the fact that the Gaussian filter half powerpoint was one hour. The moving averages of course had side lobes. It might be worthwhile to investigate more sophisticated digital filters. But the more interesting question is what time scales are really represented in hindcast data that is reported every hour but is based on three or six hour wind fields. Perhaps more work with filtering the continuous measurement time series would help answer that question.

Our response: We want to start by pointing out that the one hour value for the Gaussian filter was the standard deviation, meaning that it is expected to filter also longer time scales, since the filter also "reaches" them beyond the one standard deviation. Compared to moving averages or Fourier-filter, the Gaussian filter doesn't have a equally sharp, well defined, cut-off. This is why we also compared it to the moving average, and ensured that the filter is functioning on time scales close to what Forristal et al. (1996) recommended.

We fully agree that the issue of what time scales are actually represented (both in modelled and measured data) is by no means obvious, nor solved in this paper. For modelled data the use of the identical forcing with different intervals (i.e. every 15 minutes, every hour etc.) should probably be used, since using different products introduces other sources of uncertainty. This would also give the opportunity to see which filters (if any) can consolidate wave data generated with the different wind forcings. Here a wide variety of more advance filters should be used (as you suggested).

The data used in our study can, unfortunately, not meet the needs of a more detailed study into this subject, because of other sources of uncertainty. We therefore had to limit ourselves to raising the subject up for discussion and making a first attempt at a reasonable solution.

R1: Different users are interested in different time scales. Ship designers often want to know the three hour sea state for use in model basins. Calculating extreme values of

individual wave heights from shorter averaging times where Hs is not varying may be more accurate. Would calculations of individual wave heights from the hindcast data match those from thirty minute measurements (or chi-squared augmented hindcasts)?

Our response: Unless the lack of sampling variability is accounted for, then model and measurement data probably doesn't match for extreme values (as noted by Forristall et al. 1996 and our Fig. 3c). For mean values they should match. If sampling variability is properly accounted for, then also extreme statistics should match (assuming a perfect model etc.), but to account for this perfectly may not be possible (see also the following answer). The longer the averaging time, the less of an issue sampling variability becomes, but using an assumption of stationarity for three hours could probably be a large source of error, especially in sheltered areas and small basins. So you have to choose your poison in many cases.

R1: The difference in return periods between the chi-squared and filtered analyses deserves comment. If the chi-squared augmentation worked perfectly, wouldn't they be equal? Looking at Figure 3, it seems that the extreme wave height in the measurements has a larger deviation from the smooth curve than most of the artificial chi-squared data. That makes me think that the measurement is an outlier to the chi-squared distribution. Why don't you plot the variability of the measurements against a chi-squared distribution to check that?

Our response: If the chi-squared augmentation AND the filtering were perfect, then the results from the two data sets should agree. However, both are likely to be flawed, and this is the reason we decided to include both approaches even though they make the paper a bit more difficult to grasp. You are probably right that the maximum measured wave height is perhaps on the tail end of the distribution (which, to be fair, is not surprising for an unexpectedly high measurement). Of course, even if the chi-squared augmentation is perfect, it can only match the variability in an average sense (please also note, that panel a) and b) are different storms, since neither hindcast covered 2019).

As to plotting the variability of the measurements against a chi-squared distribution. This is a very attractive idea, but it would require us to know the "true" underlying significant wave height to relate the variation to that value (i.e. the measured values needs to be normalized by the "true" values, otherwise we are just plotting the distribution of the significant wave height, not the chi-squared distribution of the variation). One attempt to find the "true" value is the filtered time series, but we know this is not perfect. The other approach is to add variability to the model, but then we obviously assume that the variability follows a certain form. The Hs-spectra in Fig. 2 are essentially an attempt to visualize how well we are capturing the differences between the "true"/modelled and the measured/chi-squared-aumented values, even though we can never know both values in either pair.

R1: And finally, why do you think the hindcast of the recent storm was so bad?

Our response: We are not quite sure which storm this is referring to. If it refers to Fig 5, then in out opinion the hindcast was not that bad, with a quite accurate timing, although with a slight overestimation of the significant wave height. For the bias, the most obvious culprit is the wind forcing. The HARMONIE wind product is known to produce a positive bias in the modelled significant wave height. As to why this is, it is probably a part of the more general problem of wave model development, namely that we have to "tune to the mean" even though we are interested in the extremes. In other words, the physics might change in extremely high winds.

We also want to point out that several aspects of the wave field were simulated correctly, as the wave period and wave direction time series (Fib 5 b & c) show. Lastly, the accuracy of the ice product is normally also a possible source of error in this region, but this was not the case during this mild winter.

NB

Dr Jani Särkkä informed us, that while the ERA-Interim had a resolution of 3 hours, the downscaled product actually had a temporal resolution of 1 hours. Nonetheless,

it is evident from the spectra in Fig. 2 that the WAM data doesn't capture the same temporal scales as WAM forced with a wind forcing with a native temporal resolution of 1 hour (compared to Fig. 5.1, page 39 in Björkqvist, 2020). We will correct this to the text and amend the discussion to reflect what we stated above.

Nat. Hazards Earth Syst. Sci. Discuss.,
https://doi.org/10.5194/nhess-2020-190-AC2, 2020

[Figure]

R2: The manuscript that the authors presented is overall well written and discuss an interesting and imported issue of how to compare and combine model and observational data. Further the authors use this new combined dataset to investigate return values of extreme significant wave heights in the northern Baltic Sea. I generally recommend this manuscript for publication after minor revision, I would like to encourage the authors to address the following comments.

Our response: Thank you for taking the time to review our manuscript. It is greatly appreciated. We have answered your questions and comments below.

R2: Comments on the homogeneity of the new combined datasets.

The authors stated that they construct a time series of two hindcast simulation and one observational dataset. To account for the different variability in the datasets they using a low pass filter for the observation and adding variability to the model data. They analyse and discuss the different methods and the results. I would suggest that combining datasets from different hindcasts, with different atmospheric forcing and different temporal resolution should lead to some inhomogeneity. The authors stated that there is some kind of trade-off between homogeneity and preserving the "true" wave field when merging observations and model data. The manuscript would benefit to also discuss possible effects of combining different models datasets to the analysis of extreme values.

Our response: We acknowledge that using two different data sets is not ideal. Our original thought was to use the SWAN data set (1965-2005), and only fill up the five years (2006-2010) with the WAM data set. Nonetheless, we chose to also repeat the calculations with using WAM as the primary models since it quantifies (although not perfectly) the effects of the temporal resolution of the wind forcing, the differences between the models (different physical parameterizations), and data inhomogenities.

Using SWAN as the primary model is not expected to suffer fatally from inhomogenities between model data, since WAM is only used for five years. The differences between the data sets might therefore also be as much due to the difference in model data (even if both models hypothetically would cover the period up to 2010), as to the inhomogenities caused by combining two different data sets. We will add a discussion about this issue to the revised manuscript.

R2: If correctly understand, the two data series marked with WAM and SWAN only differ between the period 1979-2010 (30 out of 55 years). On one hand, the results of the return period differ quite a bit (especially using the GEV), on the other hand, some results seem to have almost no difference. Is there any explanation of these different

outcomes? Could it be the one model dominates the distribution?

Our response: 30 out of 55 years is over half of the time, so the difference is expected to be clearly visible if differences between the model data exists in the first place. We think that the most important aspect here is how the modelled extremes compare to the measured extremes. In essence, the modelled extremes are typically slightly lower in the SWAN data (see Fig. 3c), meaning that the measured maximum is more of an outlier when viewed against SWAN data (compared to if it's viewed against the WAM data).

Your observation that the GEV results differ most is important. The reason behind this is that the more parameters we have, the more risk we have of "over fitting". The curvature of the GEV distribution is determined mostly by the lower annual maxima (simply because there are more of them). A small change of the curvature can mean a large difference at the tail, where the observed maxima exists. The more exactly we fit the data (i.e. using more parameters), the more certain we have to be that the data is both reliable and homogenious. We might conclude, that if there is issues with data homogenity, the GEV/GDP distribution should be avoided because of the risks mentioned above. We will mention this in the revised manuscript.

R2: Technical comments

There is a discrepancy between the time series plotted in Figure 5, the text in section 4.2 and the colour coding in Figure 6. Whereas it is described and visible in Figure 5 that the model overestimates the observations, the coloured squares in Figure 6 indicate higher wave height for the observations (orange (8-9m) square over yellow (7-8m) area).

Our response: The discrepancy is because the time series in Fig. 5 is the original WAM-HARMONIE hindcast. The maximum wave heights in Fig. 6, again, are taken from the calibrated WAM-HARMONIE hindcast (the calibration is mentioned in the manuscript on page 12, lines 2-4). The caption in Fig. 6 mentions that is uses the

calibrated hindcast, but we will add information to the caption of Fig. 4 that this is the uncalibrated hindcast.

We acknowledge that it can be slightly confusing for the reader, but Fig. 5 is meant to show the performance of the model (and motivate the need for a calibration), while Fig. 6 is meant to show the best possible estimate of the spatially distributed significant wave height. We therefore feel that the both figures are well motivated in their current form.

R2: Also, it is stated that large parts of the Bothnian Sea show a significant wave height above 8m, but the orange to red area in Figure 6 is about a third to a quarter of the Bothnian Sea area. On the other hand, it is stated that the maximum in the Bothnian Bay is above 6m, which is confirmed by the image.

Our response: This is a language mistake. We meant to communicate that the area where 8 m was exceeded is not small. We will correct the sentence to:

"Fig. 6 shows that the significant wave height exceeded 8 m in a wide area south of the Bothnian Sea wave buoy, even reaching 9 m in the southernmost part of the basin."

R2: The titles of the four panels in Figure 7 show two times "SWAN-X2" and two times"SWAN-filtered". If this is correct it is at least confusing for the reader as I would expect filtered vs X2 and annual maxima vs POT. Also it is not mentioned which buoy is represented in the Figure 7, Bothnian Sea buoy or Finngrundet buoy.

Our response: This is correct, since we use annual maxima and POT for both the filtered and the X2 data, thus resulting in four different data sets. This means that for a AM vs. POT comparison we can compare the left column to the right column, and for a X2 vs filtered comparison we can compare the top row to the bottom row. We agree that having several data sets makes it more difficult for the reader, but we felt it was necessary to present the different options because of the significant differences different methods had on the results. We tried to alleviate the possible confusion to

consistently use crosses (x) for data with variability and circles (o) for data points without variability. Unfortunately, we found no way to simplify this further without removing essential information.

We will add the information that Fig. 7 shows data from from Bothnian Sea wave buoy.

R2: Finally, probably ":5(216)" are missing at the end of the DOI for reference Forristall et al. 1994, DOI https://doi.org/10.1061/(ASCE)0733-950X(1996)122:5(216)

Our response: Thank you for pointing this out. We will correct it.

NB

Dr Jani Särkkä informed us, that while the ERA-Interim had a resolution of 3 hours, the downscaled product actually had a temporal resolution of 1 hours. Nonetheless, it is evident from the spectra in Fig. 2 that the WAM data doesn't capture the same temporal scales as WAM forced with a wind forcing with a native temporal resolution of 1 hour (compared to Fig. 5.1, page 39 in Björkqvist, 2020). We will correct this to the text and amend the discussion to reflect what we stated above.

Editors comment: I am further asking you to extend the final discussion to the following issues. 1) Your results show clearly that the four datasets produce quite different estimates for the return time of the peak swh value on January 1st and you suggest that the Gumbel and Exponential produce the best fit. Could you please explain better the objective criterion that you use for discarding the results of the GDP and the GEV?

Our response: The processes involved with extreme value analysis are infamously subjective, and a visual analysis of the fit is often suggested (e.g. Coles, 2001). There are obviously objective tests, such as the Kolmogorov-Smirnov test, but then the subjectivity is shifted to choosing a "proper" test. We could introduce an "objective" test that match our subjective interpretation, namely that a visual analysis how the distributions fits the most extreme values of the data set and the extremely long (even infinite) return period values given by the 95% confidence interval of the shape parameters of the GEV/GPD fits.

Nonetheless, we feel that masking our subjective analysis with "objective" numbers would be highly misleading, since the test might behave differently with different data sets. In the end we think that the intended readership gets the best theoretical and practical benefits from the transparent analysis of our manuscript that weights the pros and cons of different options and solutions.

The manuscript now clearly states both that the best-fit analysis was visual (clean manuscript page 14, line 24 & page 18, line 11), and that the determination of the "most reliable" distribution was done subjectively (page 14, line 28). In addition, the detailed motivation for mistrusting the GEV/GPD-fits are stated on page 14, lines 17-23), and in section 6.3 (especially page 18, lines 12-15 & lines 27-34).

Editors comment: 2) You produce a final estimate by averaging Gumbel and exponential. It is not actually clear to me which values you use for producing the 104 years (your final best estimate). What is the methodological basis of this average? how you estimate the corresponding 95% confidence range?

Our response: The values used to produce the final best estimate is defined on page 14, lines 28-30. However, will will also mark them in italics in Table 3 and refer to this on page 14 line 23).

There is no deeper methodological basis for this average. We would argue that the insights into how to treat different types of data plays a bigger role in our study than arriving at a "final number". That said, we felt we need to provide some estimate instead or merely drowning the reader in different results and numbers, and the average is simply meant to condense the different results into a single, actionable, return period. We feel that the manuscript is balanced in how these two opposing goals are presented.

The confidence intervals are also based on a simple average of the confidence intervals of the individual results (now explicitly mentioned on page 16, lines 3-6). This is simplistic, but was motivated by two reasons: 1) We are not aware of a single methodology to combine the confidence intervals of different type of distributions, although some kind of monte carlo method might have been a suitable alternative, 2) the results clearly showed that the confidence intervals of the fits were not very relevant for quantifying the uncertainties involved, since they were both smaller or larger than the variations between different methods and/or data sets. We therefore decided that it was far more relevant to include an extensive insight into the variation of the results.

Editors comment: 3) Usually, structural planning is interested mainly in assigning return times to selected thresholds, e.g. to 7, 8, 9 10 m swh. Could you produce a table to show the return period of selected thresholds according to the different datasets and methods? Could you discuss the implications of your results for these estimates?

Our response: We have added a Table 4 with return period estimate for wave height between 6 and 9 metres. The most important distinction is that we should always consider if we are calculating return periods for measured values (with sampling variability) or modelled values (without sampling variability). This is a more important consideration that which model data set was used. We have expanded the discussion to cover this aspect.

[revised manuscript text omitted]